

# APPLICATION OF ISOTOPES AND WATER BALANCE ON LAKE DULUTI-GROUNDWATER INTERACTION, ARUSHA, TANZANIA

Nancy P. Mduma[1], Hans C. Komakech[1], Jing Zhang[2], Alfred N.N Muzuka[1]

[1]Department of Water and Environmental Science and Engineering, Nelson Mandela African Institution of Science and Technology, 23311, Tanzania

[2]State Key Laboratory of Estuarine and Coastal Research, Shanghai, 200062, China

*Correspondence to*: Alfred N.N Muzuka (alfred.muzuka@nm-aist.ac.tz)

**Abstract.** Water chemistry, and stable isotopes of oxygen and hydrogen ($^{18}O$ and $^{2}H$ respectively), were used to characterize and quantify Lake Duluti – groundwater interaction. Physico-chemical parameters: temperature, pH, electrical conductivity, dissolved oxygen, total dissolved solids, alkalinity, major cations and anions were used to determine chemical characteristics of the lake and to assess its relationship with groundwater sources. Physico-chemical parameters showed abundance of major cation and anions in the Lake water in the following order $Na>Ca>K>Mg$ and $HCO_3>Cl>F>SO_4>NO_3$. The lake water was predominantly $Na-HCO_3$ type in both wet and dry season, while spring waters were mostly of $Ca-HCO_3$ type, and boreholes were of $Ca-Na-HCO_3$ type during the dry season. In the wet season, springs and boreholes were mostly of $Na-HCO_3$ and $Na-K-HCO_3$ types respectively. Isotopic results indicate that evaporation-induced isotopic enrichment prevails in the lake and contributes significantly to water loss from the lake. The $\delta^{18}O$ of the lake water averaged +6.1‰ while that of well/boreholes and springs averaged -1.2‰ and -2.1‰ respectively. Similarly, the δD of lake water averaged +24.2‰ while that of well/boreholes and springs averaged -12.9‰ and -12.2‰ respectively. Stable isotope calculations indicate that the lake loses it water to the groundwater aquifer. Water balance model equations used to quantify the level of lake - groundwater exchange found that the Lake Duluti receives and recharges more groundwater than it receives from precipitation and surface runoff. Groundwater thus plays a major role in the hydrological system of Lake Duluti. The findings in this research are of assistance to policy makers and management personnel to make use of the information provided for better management of the lake water.

Keywords: Water balance, Isotopes, Crater Lake, groundwater, hydrochemistry



## 1 Introduction

All freshwater bodies including rivers and lakes constitute a valuable resource to the world. In Tanzania, more than 75% of the population depends on surface water (Kashaigili, 2012; Elisante and Muzuka, 2015). Exploitation of surface water is on the increase due to population increase and its associated consumption for domestic, industrial, agricultural, and other purposes/uses linked to improvement of standard of living. Quantity and quality of surface water in many parts of the world are threatened by natural and anthropogenic activities. In Tanzania, the quality and quantity of surface water are deteriorating due to human and natural activities (Elisante and Muzuka, 2015) and consequently threatening human health and environment in general. Due to such deterioration, a portion of the population uses groundwater as an alternative water source. Nevertheless, the hydrological part of surface and groundwater sources and their relations in many areas of Tanzania remain poorly understood. In order to conserve and utilize the valuable water resources effectively and be able to exploit it sustainably, it is important to consider hydrological systems, which implicitly make it essential to understand the hydro-geochemical characteristics of lake and groundwater sources (Kumar, *et al*., 2006).Various uses of water e.g. for domestic, irrigation or industrial purposes depend on its quality particularly physico-chemical and biological quality. Water quality and quantity are controlled by a number of factors, such as climate, geochemistry, duration of contact with rocks, topography of the area, saline water intrusion in coastal areas, human activities on the ground, etc., (Reghunath *et al.*, 2002; Hiscock, 2005; Ayenew, 2006; Martinez *et al.*, 2015). Apart from these factors, the interaction between surface water and adjacent groundwater and the mixing of different types of groundwater may also play important roles in determining quality of surface and groundwater (Darling *et al.*, 1996; Reghunath *et al.*, 2002). Lake Duluti, a crater lake on the slopes of Mt. Meru, is a topographically closed freshwater body with no obvious surface inflow and outflow and therefore, assumed to receive and discharge its water through subsurface flow, precipitation and evaporation (Öberg *et al.*, 2012). Evaporation in closed lakes may result in accumulation of solutes (Belay, 2009; Mckenzie *et al.*, 2010; Öberg *et al.*, 2012), which may compromise its potential to be used for human consumption. According to Öberg *et al.*(2012), Lake Duluti loses its solutes through subsurface flow. Fresh water lakes and aquifers are important water resources and are used extensively for various human activities such as agriculture, industrial and domestic consumption (Yihdego and Becht, 2013; Mulwa *et al.*, 2013). Lake Duluti and surrounding aquifers are important water resources in the area and are used for irrigation and domestic water supplies. Additionally, there is a plan to use water from the lake in future as a source of water for Arusha city. Nevertheless, continued abstraction from the lake and the surrounding aquifers, which is on the increase, may have potentially negative impact on water levels and quality of these resources. Hydro-chemical data with other information such as water level, stable environmental isotopes can be used to ascertain the interaction between surface water and groundwater (Mckenzie *et al.*, 2010; Kamtchueng *et al.*, 2014; Martinez *et al.*, 2015). However, prior to this study, such information for L. Duluti was not available. Conducting an estimate of Lake Duluti water balance provides a comprehensive understanding of the water flow




system and water resources in the study area. This study has addressed the role of groundwater, with a focus on quantifying
the groundwater exchange with Lake Duluti. The findings from this research will help to provide baseline information about
the lake hydrological system for effective management of the lake. The results will be used as evaluation tools for
management of lake water and reveals the effects of surrounding groundwater sources on the survival of the lake.
**1.1 Description of the study area**
The study area, which include Lake Duluti, and its surroundings, is located east of Arusha town, between latitude 3°21′ S to
3°25′ S and longitude 36°46′ E to 36°49′ E in northern Tanzania as in Fig.1. The lake is situated at an altitude of 1290 m and
has an estimated surface area of 0.6 km$^2$ and maximum depth of approximately 9 m (Öberg *et al.*, 2012). The catchment is
confined by the crater walls with no surface inlets or outlets. Additionally, unlike many other crater lakes, Lake Duluti water
is reported to have low salinity (Öberg *et al.*, 2012), indicating a continuous exchange of water masses and loss of ions
through the process. It is situated in one of the parasitic volcanoes that were associated with Mt. Meru eruptions.
Geologically, Lake Duluti has a crater rim primarily consisting of finely stratified ashes and tuffs. Elevated ridge to the
North, East and West of the lake consists of weathered vesicular basalt. Lake Duluti and surrounding sampled areas are
dominated by alkaline igneous rocks, mostly phonolites and nephelinites (Wilkinson *et al.*, 1983; Dawson, 1992; Öberg *et*
*al.*, 2012) as in Fig 1 of the geology map of the study area, and that of sampled water points. The dominant climate is
tropical type with clearly defined rainy and dry seasons. Rainfall is bimodal with an annual average above 1000 mm/year
distributed mainly between two seasons namely, short rains of October/November to January and long rains of
February/March to May (Öberg *et al.*, 2012; Meru District Council, 2013) .The decrease and high variability of rainfall
contributes to the demand of water for different activities taking place in the surrounding areas of L. Duluti. Lake water
levels records from (2003 to 2014) showed top water level having slight fluctuations through a range of about 1.5metres.

<<Figure 0>>
**2 Methodology and Theoretical Considerations**
Hydro chemical data and Stable environmental isotopes ($^2$H and $^{18}$O) have been used to characterize and perform water
balance of Lake Duluti. Fieldwork involved collection of water samples in pre-washed polyethylene bottles from Lake
Duluti vertically and laterally, and surrounding groundwater resources during dry (January-February 2015) and wet (March –
April, 2015) seasons. During the water sampling exercise for determining chemical characteristics, each bottle was rinsed at
least three times with the water to be collected before sampling. Dissolved oxygen (DO), pH, temperature, total dissolved



solids (TDS) and electrical conductivity (EC) were measured on site using a multi-parameter meter. Water samples from
Lake Duluti were collected at ten sampling stations, and at different depths including surface and bottom using a water
sampler. Water samples from boreholes were collected after pumping to allow substitution of stagnant water by freshwater
from aquifer. Water samples were also collected from surrounding springs. Samples for cation analysis were acidified with
nitric acid (Supra pure) onsite to pH<2 to prevent precipitation of cations and water sample for nitrate were acidified with
sulphuric acid to pH<2. Finally samples were placed in clean ice boxes, and transported back to the laboratory where they
were stored in the refrigerator until analysis. The geographic co-ordinates and altitude were also taken at each sampling
location using a Geographic Positioning System (GPS). Sampling of water from Lake Duluti and surrounding ground water
sources during dry and wet seasons and at points/site mention above was carried out for isotope analysis. Water sampling
was done as described above on sampling for chemical characteristics. However, in order to minimize isotopic fractionation
resulting from evaporation, samples were kept in Amber glass bottles with Teflon-lined caps and stored in the refrigerator at
at -4°C in the Nelson Mandela African Institution of Science and Technology (NM-AIST) laboratory prior to analysis at
State Key Laboratory of Estuarine and Coastal Research, China. Determination of major cations ($Ca^{2+}$, $Mg^{2+}$) was done by
titration, while $K^+$ was determined by using COD Multi-paremeter spectrophotometer (HANNA® 83099) at the (NM-AIST)
water quality laboratory. Sodium was analyzed at Seliani Agricultural and Research Institute (SARI) by the Flame
Photometer method. Major anions ($CO_3^{2-}$, $HCO_3^-$, $Cl^-$,) were determined by titrimetric method while $NO_3^-$ and $SO_4^{2-}$ were
determined by COD Multi-parameter at NM-AIST water quality laboratory. The stable isotopes compositions of oxygen and
hydrogen were determined using High temperature conversion elemental analyzer-mass spectrometry (TC/EA-MS) by
injecting 0.1 ml of the sample using a microliter syringe. Helium was used as carrier gas of the sample through a heated
septum. $H_2O$ samples were reduced by reaction with glassy carbon and vaporized at high temperature of 1400°C. All
samples were measured at least four times and the final result was reported as mean value. The results are reported in **δ-**
notation in per mil (‰) and reproducibility was better than 0.3 and 3‰ for $\delta^{18}O$ and $\delta D$, respectively. STATISTICA$^{TM}$
StatSoft 7.0 was used to carry out all statistical tests. Average, standard deviation, maximum and minimum value were
calculated for all parameters in dry and wet season. Multivariate analysis of variance was used to assess the effect of water
sources on water chemistry while the interrelationships among different variables were assessed using correlation matrix.
Water balance and isotopic balance equations were used collectively to ascertain the contribution of groundwater to the
hydrological state of L. Duluti. Water budget equation for a closed lake can be estimated using the  Sacks, Swancar et al.,
(1998) equation as shown in Eq. 1.

$$\frac{dV}{dt} = Gi + P + Si - Go - E - So = 0 \tag{1}$$

Where;



V= Volume of water in the lake, t= Time, P= Precipitation, E= Evaporation, Gi= groundwater inflow to the lake, Go= Lake
outflow to the ground, Si = surface water inflow and So = Surface water outflow. Equation (1) can also be written as:

$$Gi + P + Si = Go + E + So \qquad (2)$$

Taking into consideration of isotope mass balance of $\delta^{18}O$ and $\delta^2H$ for a closed lake (Krabbenhoft, Bowser $et~al.$, 1994), Eq.
(2) can be expressed as shown in Eq. (3) and Eq. (4).

$$\delta_{gi}Gi + \delta_P P + \delta_P KP = \delta_{go}Go + \delta_E E + \delta_L So \quad \text{for } \delta^{18}O \qquad (3)$$

$$\delta_{gi}Gi + \delta_P P + \delta_P KP = \delta_{go}Go + \delta_E E + \delta_L So \quad \text{for } \delta^2H \qquad (4)$$

Where;
KP= total rainfall that is converted to runoff, K= Runoff coefficient (estimated based on field slope (Ayenew, 1998)),
$\delta_L$=Isotopic composition of lake water, $\delta_{gi}$ = Isotopic composition of groundwater inflow to the lake, $\delta_P$=Isotopic composition
of precipitation, $\delta_E$=Isotopic composition of evaporation, $\delta_{go}$=Isotopic composition of lake water outflow to the ground.
Slope of the area surrounding the lake has been determined from Digital Elevation Model using ArcGIS 10.2 and the side of
possible runoff has been determined by SWAT. Lake Duluti, a crater lake with its catchment confined by the crater walls,
with no surface inlets or outlets, it is assumed to have negligible or no surface runoff and hence $S_o$ = 0. For water under
Isotopic steady state condition, isotopic composition of water recharging the ground from the lake is the same as that of the
lake (Kebede, Ayenew $et~al.$, 2004) (**$\delta_{go}= \delta_L$** ).
From the theories and assumptions above, Eq. (3) and Eq. (4) become;

$$\delta_{gi}Gi + \delta_P P + \delta_P KP = \delta_L Go + \delta_E E \quad \text{for } \delta^{18}O \qquad (5)$$

$$\delta_{gi}Gi + \delta_P P + \delta_P KP = \delta_L Go + \delta_E E \quad \text{for } \delta^2H \qquad (6)$$

Isotopic composition of evaporated water is derived from the assumption that, isotopic composition of the lake is the sum of
isotopic composition of groundwater inflow to the lake, precipitation and evaporation.

$$\delta_{gi} + \delta_P + \delta_E = \delta_L \qquad (7)$$

Hence,

$$\delta_E = \delta_L - \delta_{gi} - \delta_P \qquad (8)$$

Final Equations for L. Duluti Isotopic water balance;

$$\delta_{gi}Gi + \delta_P P + \delta_P KP = \delta_L Go + (\delta_L - \delta_{gi} - \delta_P)E \quad \text{for } \delta^{18}O \qquad (9)$$

$$\delta_{gi}Gi + \delta_P P + \delta_P KP = \delta_L Go + (\delta_L - \delta_{gi} - \delta_P)E \quad \text{for } \delta^2H \quad (10)$$

Two component mixing equations have been used to ascertain the mixing of lake water and groundwater and to estimate the
relative contribution.

$$1 = f_L + f_G \qquad (10)$$



Where; $\delta_{Ls}$ = Isotopic composition of lake surface water, $\delta_{G}$ = Isotopic composition of groundwater, $f_{L}$ = Fraction of lake
water, $f_{G}$ = Fraction of groundwater

$$\delta_{mix} = \delta_{Ls}f_{L} + \delta_{G}f_{G} \qquad (11)$$

But $1 - f_{L} = f_{G}$ and hence;

$$\delta_{mix} = \delta_{Ls}f_{L} + \delta_{G}(1 - f_{L}) \qquad (12)$$

$$\delta_{mix} - \delta_{G} = \delta_{Ls}f_{L} - \delta_{G}f_{L}$$

$$f_{L} = \frac{\delta mix - \delta G}{(\delta Ls - \delta G)} \qquad (13)$$
$\delta_{mix}$ is assumed to be the isotopic composition of lake water minus that of precipitation, groundwater and evaporation, as its
proved by water balance calculation that the lake is recharged by groundwater.
Therefore the fraction of groundwater, $f_{G} = 1 - f_{L}$

**3 Results and Discussion**
**3.1 Physico-chemical characteristics of Lake Duluti and surrounding groundwater sources**
Generally, the physicochemical characteristics of Lake Duluti water and surrounding groundwater varied markedly with
seasons, depth and locations. Range, mean and standard deviation values for physicochemical parameters from Lake Duluti
and surrounding groundwater sources are presented in Table 1.

<<Table 1>>

In the study conducted to assess the physic-chemical characteristics of Lake Duluti and its relation to groundwater sources,
pH of the Lake, which was found to be mildly alkaline (pH>7) may have been a result of natural processes including
contribution by surrounding alkaline rocks (Wilkinson *et al.*, 1986; Öberg *et al*., 2012). Similar observation was reported by
Öberg *et al*., (2012). The observed decreasing trend of pH was probably due to decomposition of organic matter and
respiration occurring in the lower water levels. Slightly acidic to slightly alkaline values were recorded for groundwater
sources (springs and boreholes) in Meru district, which have also been reported by Elisante and Muzuka, (2015). Water
temperature fluctuations occurred between different points of the Lake water column due to response to incomplete mixing
or homogenization by wind leading to stratification (Fig. 3c and d). Thermal stratification in Lake Duluti occurs at a depth of
2 m from the surface where the top 2 m are well mixed and aerated while poor mixing occurs at the depth below 2 m. Similar
observation was observed in this study where the top 2 m of lake water showed higher temperature than the lower levels
(Öberg *et al*., 2012). Due to low temperatures recorded in the bottom water, such water was denser than the top water a
likely cause of poor mixing of the lake water. Poor mixing affected the distribution of DO and pH within the Lake. Lower





temperatures at the bottom of the lake could also be an indication of groundwater input or flux to the lake. This is supported
by the results of oxygen isotopes which showed a general decreasing trend in isotopic composition with depth. DO decreased
significantly with increasing depth during the dry season (r=-0.68, p<0.01) and (r=-0.80,p<0.01) in wet season probably due
to the fact that, wind overturning is not enough to cover the water column. The EC and TDS values in the lake water and
groundwater, which showed no significant differences, were likely to be a result of inlet and outlet of lake water through
groundwater.

<<Figure 2>>

Sodium, which was the dominant cation in the lake water, was likely to be a result of silicate weathering and/or dissolution
of soil salts discharged into lake water through groundwater (Subbarao *et al.*, 1996; Mamatha and Rao, 2010; Rao *et al.*,
2012). However, it has been observed that the rate of accumulation of some ions like $Na^+$ in the lake water is very low due to
loss of ions to groundwater; this has also been reported earlier that Lake Duluti loses some ions through the groundwater
(Öberg *et al.*, 2012). Furthermore, ion exchange influences the higher contribution of $Na^+$ than that contributed by $Ca^{2+}$. The
concentration of $Ca^{2+}$ in the lake water, which was significantly lower than in the surrounding springs and boreholes was
likely to be a result of precipitation as $CaCO_3$ due to high concentration of $HCO_3^-$ and $CO_3^{2-}$ in the lake water (Anazawa,
2001 ). Lake Duluti was observed to have a higher concentration of $K^+$ as compared to the groundwater sources probably due
to cumulative effect as water comes from the ground. The observed higher concentration of fluoride in lake water than the
surrounding groundwater was probably a result of accumulation due to evapo-transpiration. It is very likely that, the lake
receives water from the ground with low concentration of $F^-$, which later accumulates as the water evaporates. In addition,
the dominance of $Na^+$ and $HCO_3^-$ ions in the lake indicates precipitation of $Ca^{2+}$ as $CaCO_3$. Few groundwater sources around
the Lake had relatively high fluoride concentration a likely result of interaction of water with the volcanic rocks rich in F-
and alkali (Ghiglieri *et al.*, 2010).

<<Figure 3>>

The concentration of $Cl^-$ was relatively high in the Lake water than groundwater during the two seasons although the
difference was not significant. Also, there was no significant variation of $Cl^-$ concentration with seasons. Probably, the main
sources of $Cl^-$ in the Lake and surrounding groundwater sources is dissolution of salt deposits and weathering The observed
significantly high concentration of carbonate in the lake water than in surrounding springs and boreholes was likely as a
function of dissolved carbon dioxide, temperature, pH, cations and other dissolved salts. Groundwater sources were observed



to have higher concentration of $SO_4^{2-}$ than the concentration observed in the lake. This may be contributed to dissolution of
salts deposits and contamination caused by different human activities such as agricultural activities and livestock keeping.
The concentration of $NO_3^-$ in lake water was significantly lower (p<0.01) in lake water than in surrounding groundwater.
This could be attributed to either minimum anthropogenic input or high primary productivity of macrophytes and
phytoplankton. During fieldwork, it was noted that a small part of shallow lake areas have been colonized by macrophytes
vegetation such as papaylus. Such plants have been used in constructed wetland to reduce nutrient loading in waste water
(Gottschall, 2007). Elevated nitrate concentration (above 10 mg/L) was observed in some of the surrounding groundwater
sources indicating pollution due to agriculture and sanitation facilities. Similar elevated nitrate concentration due to
contamination with sewage and animal manure in groundwater source in Meru district have been reported by Elisante and
Muzuka, (2015).

**3.1.2 Hydro-geochemical Facies**
The Hydro-chemical water type of water samples from Lake Duluti and surrounding groundwater sources in dry and wet
seasons from the study area is represented in Piper diagram in Fig 4 for the dry season and in Fig 5 below for the wet season,
where concentration is assigned in % meq/L.

<<Figure 4>>

The hydrochemistry of the sampled water shows that there is a clear distinction between the lake, spring, and boreholes
samples collected in the studied area. This is clearly shown in a Piper diagram by plotting the major ions of water chemistry
on a single four sided diagram. The Piper diagram describes the composition of water and classifies it into ionic type based
on the dominant cation and anions. Plot of Lake water results on the diamond diagrams were observed to fall within the
alkali metals than alkali earth metals (Na +K >Ca + Mg). Weak acid anions were dominant over strong acid anions ($HCO_3$>
$SO_4$ + Cl). Therefore, based on this classification, the water types found in the study area with respect to the water sources
are Na-$HCO_3$ type for Lake Duluti, Ca-$HCO_3$ types for springs and Ca-Na-$HCO_3$boreholes during dry season and Na-$HCO_3$
type for Lake Duluti, Na-$HCO_3$ and Na-K-$HCO_3$ types for springs and boreholes, during wet season respectively.

<<Figure 5>>

**3.1.3 Interrelation of Chemical Parameters**



Correlation Matrix was used to evaluate the interrelationships among physico-chemical parameters of Lake Duluti water and
surrounding groundwater sources. Correlations among physico-chemical parameters in Lake Duluti water are presented in
Tables 2 and 3 while for surrounding groundwater are presented in Tables 4 and 5. The pH showed significant negative
correlation with EC, TDS, $HCO_3^-$ (r=-0.69,-0.76,-0.63, p≤0.05), and significantly positive correlation with temperature,
$CO_3^{2-}$ (p≤0.01, r = 0.51, 0.94, respectively) during dry and wet seasons. Also, pH showed significant positive correlation
with $Na^+$ and $Ca^{2+}$ (r=0.58, 0.61, p≤0.05) during the dry season, this shows that, the Lake water is characterized by alkaline
surroundings, which tends to dissolve aquifer minerals such as fluoride-bearing minerals. The EC and TDS showed
significant positive correlation with temperature, $HCO_3^-$ at (p≤0.05, r=0.55, 0.57), and negative correlation with $CO_3^{2-}$ at
(p≤0.05, r=-0.65, -0.72) during dry and wet season indicates that, the Lake is mainly controlled by $HCO_3^-$ ions, which
depend upon respiration of aquatic organisms, decomposition of organic matter and mineral solubility. $Na^+$ in the Lake water
showed significant positive relationship with $K^+$ (p≤0.01, r=0.65), $Ca^{2+}$, $Cl^-$, $CO_3^{2-}$ at (p≤0.05, r=0.53, 0.60, 0.58) and
negative correlation with $F^-$ during dry and wet seasons, indicates the influence of evaporation and dissolution of minerals.
$K^+$ was also observed to have significant positive correlation with $Ca^{2+}$ (p≤0.05, r=0.51), $Mg^{2+}$ (p≤0.01, r=0.67) indicating
mineral dissolution. Significant positive correlation of $K^+$ with $Ca^{2+}$ indicates mineral dissolution and $SO_4^{2-}$ showed positive
correlation with $NO_3^-$ at 95% confidence level indicating oxidation of organic matter. The results also show there is local
system of ground-water flow, as recharge and discharge areas (sampled sites) are adjacent to each other.

**3.2 Lake Duluti Isotopic Water Balance**
Isotope analysis results of the sampled lake, rainfall and groundwater are presented in Tables 6. Sampled water bodies for
$\delta^{18}O$ Vs $\delta^2H$ are plotted with respect to the Local Meteoric Water Line (LMWL) as shown in figure 6. Local Meteoric Water
Line (LMWL) data were obtained from the International Atomic Energy Agency (IAEA) for a for different areas of
Tanzania.The Global Meteoric Water Line (GMWL) was generated based on the worldwide scale relationship revised by
(Rozanski, 1993) with respect to the sampled isotopic composition data and relative to Standard Mean Ocean Water
(SMOW). The points clustered around the GMWL show significant rainfall recharge.

<<Figure 6>>

Isotopic compositions of sampled lake water lies to the upper right of the LMWL in Figure 6 indicating high rate of
evaporation. Water samples plotted close to the LMWL indicate that, there is limited evaporation of water (Yuan, 2011 ).
Water falling outside the LMWL indicates the presence of evaporation; this is due to the fact that evaporation-induced
isotopic fractionation and cause deuterium enrichment in water vapour and subsequently an isotopic enrichment of $^{18}O/^{16}O$



relative to D/H in lake water Evaporation effects and variation of temperature of precipitation contributed to some variability
among samples. Groundwater sources appearing concentrated to the left of local meteoric water line (LMWL) are indicative
of no evaporation occurrence during recharge either at the surface or within the soil zone. The range of δ-values in Fig. 6
decreases from precipitation to lake water to groundwater.  The isotopic composition of each of the water types and their
significance is indicative of groundwater- surface water interaction as discussed below.

<<Table 2>>

The results above (Table 6) are in **δ** notation in per mil (‰) versus SMOW; reproducibility is better than 0.3 and 3‰ for
$\delta^{18}$O and δD, as per the standard sample with respect to the water samples analysed. Lake surface temperature, evaporation
of lake water and precipitation were considered to be homogenous due to climatic and environmental conditions of the
sampled area. An annual evaporation (E) of 1,700 mm/year was recorded at Arusha Airport, some 17 km west of Lake
Duluti. The average annual precipitation (P) of approximately 1,012 mm/yr was recorded at Tengeru, about 1 km from Lake
Duluti, (Tanzania Meteorological Agency). Lake average surface temperature (T) during sampling was 25.08 °C, with small
variations with depth and seasons.

The isotopic compositions of oxygen and hydrogen ($\delta^{18}$O$_L$ and $\delta^2$H$_L$) for lake water ranged from 3.5 to 6.9‰ for $\delta^{18}$O and
14.5 to 28.0‰ for $\delta^2$H. It averaged 6.1‰ for $\delta^{18}$O, 24.2‰ for $\delta^2$H in dry season and 5.3‰ for $\delta^{18}$O, 21.1‰ for $\delta^2$H for wet
season respectively (Figure 6, Table 6). Similarly, the isotopic compositions of these two parameter for rainfall averaged -
1.2‰ for $\delta^{18}$O$_P$ and -1.6‰ for $\delta^2$H$_P$ Furthermore, the isotopic compositions for oxygen and hydrogen for surrounding
groundwater sources  ($\delta^{18}$O$_G$ and $\delta^2$H$_G$) averaged -1.4‰ for $\delta^{18}$O, 12.4‰ for $\delta^2$H in dry season  and -1.9‰ for $\delta^{18}$O, -10.9‰
for $\delta^2$H for wet season, respectively (Tables 1 and 2).

Stable isotope water balance calculations suggest that Lake. Duluti loses water to the aquifer and it is more recharged by the
groundwater relative to precipitation and surface runoff. Groundwater inflow to the lake is approximately 2,430,960m$^3$/yr
while lake water discharge to groundwater is 2,902,620m$^3$/yr. The lake is recharged through precipitation by 612,000m$^3$/yr.
Hence, groundwater plays a major role in the hydrological system of Lake Duluti. Groundwater sources are less enriched
than the lake and their isotopic composition in the study area ranged from -0.5 to -2.8‰ for $\delta^{18}$O and -3.7 to -6.7‰ for $\delta^2$H.
Results show some variation in isotopic composition of sampled groundwater, probably due to differences in sources
(springs, boreholes), different sampled locations and temporal variations of recharge. There is little or no evaporation in
groundwater as these are recharged through faults/ joints as compared to the lake, because its surface water is exposed to the



atmosphere. Sampled rainfall close to the lake had isotopic composition of -1.2‰ for $\delta^{18}O$ and -1.6‰ for $\delta^2H$. Results also
shows that isotopic composition of water samples from upstream and downstream did not show any specific trend suggesting
that the water sources recharging the lake is not from a specific elevations. There is a slightly variation in the oxygen
isotopic composition of Lake water with seasons and depth indicating that the lake is to some extent not isotopically mixed
(Fig.7) . Most of the lake sampled sites showed decrease of oxygen isotopic composition with depth as evaporation on the
lake surface leads to increase in $\delta^{18}O$. Evaporation from the lake influence surface water to be more isotopically enriched.
Two component mixing equations have been used to ascertain the mixing of lake water and groundwater and to estimate the
relative contribution.The fraction of groundwater is 0.73, which shows that groundwater contributes to 73% of lake water.
This shows that groundwater play a role in lake Duluti, and it does have significant difference with the water balance results
which showed groundwater contributes 80% of lake water.

<<Figure 7>>

**4 Conclusion**
The study was done to ascertain hydro-chemical characteristics of Lake Duluti and to assess lake-groundwater interactions.
Hydro-chemistry data reveal that, Lake Duluti water is generally alkaline characterized by pH values between 8.6 and 9.3.
Alkalinity and hardness were low indicating freshness of the lake water.  Low EC and TDS reflected low concentration of
TDS probably due to limited interaction between lake water and rocks and lake water is moderately hard, ranging from 60-
120mg/l (EPA). The abundance of major cation and anions in lake water is in the following order Na>Ca>K>Mg and
$HCO_3$>Cl>F>$SO_4$>$NO_3$. In all lake water samples taken during dry and wet seasons, $Na^+$ was the prevalent cation while
$HCO_3^{2-}$ was the prevalent anion. $Na^+$ is accumulated due to evaporation, and there is lake-groundwater interaction influenced
by ion exchange and mineral dissolution. The concentrations of ions into the lake water are influenced by ion exchange,
mineral dissolution and anthropogenic activities. The dominant hydro-chemical facies is Na-$HCO_3$ water type for all water
samples in Lake Duluti, while Ca-$HCO_3$ and Ca-Na-$HCO_3$ type were present in springs and boreholes, respectively during
dry and wet seasons. The facies are caused by the influence of geogenic factors including ion exchange, mineral solubility;
mineral dissolution and evaporation. The isotopic compositions of sampled lake water indicated high rate of evaporation.
Water samples from boreholes indicated that, there is limited evaporation of water as they are recharged through faults/
joints as compared to the lake, as its surface water is exposed to the atmosphere. However, results of the water balance
calculation showed that the net ground-water outflow occurs from the lake and groundwater plays significant role in the
hydrological state of the lake. Isotopic composition of water sampled showed that the lake is more enriched than
groundwater sources and rainfall, due to evaporation and the amount of groundwater outflow probably is accountable for the





difference in the lake water salinity. Based on these findings groundwater plays a major role on the recharge of Lake Duluti
and therefore remain as base information for citing of boreholes in the study area.

**Authors Contribution**

Nancy Mduma did the field work, laboratory analysis and preparation of the manuscript. All activities related to research
design, implementation analysis and preparation of the manuscript have been supervised by Hans C Komakech and Alfred
Muzuka.

**Acknowledgement**

This research was funded by the Government of Tanzania through the Nelson Mandela African Institution of Science and
Technology (NM-AIST). VLIR-UOS IUC Programme at NM-AIST supported field activities and East China Normal
University supported the Isotope analysis of water samples.

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






**Table 1: Descriptive statistics of physico-chemical parameters of water from Lake Duluti.**

| Parameters | Dry Season | | | Wet Season | | |
|---|---|---|---|---|---|---|
| | **Max** | **Min** | **Mean** | **Max** | **Min** | **Mean** |
| **Lake Duluti** | | | | | | |
| pH | 8.93 | 8 | 8.6±0.3 | 9.8 | 8.7 | 9.3±0.3 |
| EC (µS/cm) | 570 | 524 | 547.6±14.1 | 646 | 510 | 530±31.0 |
| TDS (ppm) | 285 | 262 | 273.4±6.6 | 323 | 255 | 264.3±15.6 |
| Temp(°C ) | 27.7 | 24.6 | 25.8±1.0 | 27.8 | 25 | 25.8±0.9 |
| Na (mg/l) | 42.4 | 41 | 41.7±0.3 | 42.4 | 34.8 | 39.5±2.3 |
| K (mg/l) | 8 | 5.2 | 6.3±0.9 | 8.3 | 4.3 | 6.1±1.2 |
| Ca (mg/l) | 18 | 16.4 | 17.1±0.4 | 19.2 | 12 | 16.8±1.9 |
| Mg (mg/l) | 7.7 | 6.3 | 7.0±0.4 | 8.5 | 2.4 | 5.9±1.8 |
| Cl (mg/l) | 14.5 | 8.5 | 10.6±1.7 | 10.5 | 7.5 | 8.7±0.7 |
| $SO_4$ (mg/l) | 5 | 1 | 1.7±1.0 | 8 | 0 | 2.4±2.2 |
| $NO_3$ (mg/l) | 2 | 0.8 | 1.2±0.4 | 2.4 | 1.2 | 1.6±0.3 |
| $HCO_3$ (mg/l) | 167.2 | 134.7 | 150.5±10.0 | 155.4 | 81.6 | 117.0±20.1 |
| $CO_3$ (mg/l) | 11.3 | 1.4 | 6.9±3.1 | 34.8 | 7.3 | 21.3±8.1 |
| F (mg/l) | 3.3 | 3 | 3.1±0.1 | 2.6 | 2.3 | 2.5±0.1 |
| **Springs** | | | | | | |
| pH | 7 | 6.5 | 6.8±0.3 | 7.6 | 7 | 7.3±0.2 |
| EC (µS/cm) | 549 | 377 | 436.7±97.3 | 565 | 196 | 360.8±139.3 |
| TDS (ppm) | 274 | 189 | 218.3±48.2 | 282 | 98 | 180.2±69.5 |
| Na (mg/l) | 19 | 10 | 14.0±4.6 | 18.3 | 5.2 | 11.8±4.7 |
| K (mg/l) | 4.1 | 1.6 | 3.0±1.3 | 4.3 | 1.7 | 3.0±1.2 |
| Ca (mg/l) | 32.1 | 24 | 27.8±4.1 | 31.6 | 15.5 | 23.6±6.1 |
| Mg (mg/l) | 7.9 | 4.8 | 6.5±1.6 | 7.3 | 3.4 | 5.0±1.8 |
| Cl (mg/l) | 13.5 | 2.5 | 8.8±5.7 | 16 | 0.5 | 6.6±6.7 |
| $SO_4$ (mg/l) | 9 | 6 | 7.3±1.5 | 14 | 4 | 9.2±3.9 |





| | | | | | | |
|---|---|---|---|---|---|---|
| NO$_3$ (mg/l) | 1.6 | 0.6 | 1.2±0.5 | 4.4 | 1.6 | 2.8±1.2 |
| HCO$_3$ (mg/l) | 121.2 | 87.3 | 102.4±17.3 | 120.8 | 64 | 94.5±20.6 |
| CO3 (mg/l) | 0.1 | 0 | 0.1±0.0 | 0.4 | 0.1 | 0.2±0.2 |
| F (mg/l) | 1.6 | 1.5 | 1.5±0.1 | 1.2 | 0.4 | 0.9±0.3 |
| **Boreholes** | | | | | | |
| pH | 8.5 | 6.8 | 7.2±0.5 | 8.5 | 7.1 | 7.7±0.5 |
| EC (µS/cm) | 885 | 337 | 549.2±201.8 | 888 | 376 | 563.9±187.1 |
| TDS (ppm) | 442 | 169 | 274.8±100.8 | 444 | 188 | 282.3±93.4 |
| Na (mg/l) | 28.1 | 16.2 | 19.2±3.6 | 18 | 15.4 | 16.9±0.9 |
| K (mg/l) | 5.8 | 1.5 | 3.1±1.2 | 6.1 | 1.7 | 3.2±1.3 |
| Ca (mg/l) | 24.2 | 19.1 | 22.4±1.5 | 23.7 | 18.3 | 21.8±1.5 |
| Mg (mg/l) | 10.4 | 6 | 7.3±1.4 | 9.8 | 5.2 | 6.6±1.4 |
| Cl (mg/l) | 16.5 | 2 | 8.7±5.0 | 16 | 2 | 8.7±4.8 |
| SO$_4$ (mg/l) | 9 | 0 | 5.2±3.6 | 10 | 6 | 7.9±1.3 |
| NO$_3$ (mg/l) | 6.1 | 0.4 | 2.8±1.6 | 4.2 | 1.3 | 3.2±0.8 |
| HCO$_3$ (mg/l) | 252.6 | 84.5 | 140.5±64.3 | 130.2 | 97.8 | 106.0±10.1 |
| CO$_3$ (mg/l) | 6.7 | 0.1 | 0.9±2.2 | 2.9 | 0.2 | 0.9±1.0 |
| F (mg/l) | 4.1 | 1.3 | 2.2±1.2 | 4 | 0.8 | 1.7±1.2 |



**Table 2: Stable isotopes compositions of oxygen and hydrogen of water for lake, groundwater and rainfall.**

| No. | Sample No. | Sample Site | EC(µS/cm) | δ$^{18}$O | δ D |
|---|---|---|---|---|---|
| | | **Lake Duluti** | | | |
| 1 | LD1-S | Lake Pt A, surface | 646 | 5.4 | 19.9 |
| 2 | LD1-1 | Lake Pt A, 1m | 516 | 4.7 | 23.6 |
| 3 | LD1-2 | Lake Pt A, 2m | 520 | 4.9 | 19.4 |
| 4 | LD1-3 | Lake Pt A, 3m | 525 | 3.9 | 19.5 |
| 5 | LD2-S | Lake Pt B, surface | 541 | 5.1 | 21.3 |
| 6 | LD2-1 | Lake Pt B, 1m | 511 | 4.8 | 14.5 |
| 7 | LD2-2 | Lake Pt B, 2m | 515 | 4.8 | 18.3 |





| 8 | LD3-S | Lake Pt C, surface | 511 | 4.8 | 21.7 |
| 9 | LD3-1 | Lake Pt C, 1m | 511 | 4.8 | 23.3 |
| 10 | LD3-2 | Lake Pt C, 2m | 512 | 4.2 | 21.7 |
| 11 | LD3-3 | Lake Pt C, 3m | 520 | 3.5 | 20.5 |
| 12 | LD3-4 | Lake Pt C, 4m | 527 | 5.5 | 23.7 |
| 13 | LD3-5 | Lake Pt C, 5m | 551 | 5.0 | 17.2 |
| 14 | LD3-6 | Lake Pt C, 6m | 568 | 4.8 | 21.6 |
| 15 | LD3-7 | Lake Pt C, 7m | 560 | 4.8 | 20.6 |
| 16 | LD4-S | Lake Pt D, Surface | 510 | 5.3 | 25.6 |
| 17 | LD4-1 | Lake Pt D, 1m | 511 | 5.6 | 20.6 |
| 18 | LD4-2 | Lake Pt D, 2m | 525 | 5.7 | 21.3 |
| 19 | LD5-S | Lake Pt E, surface | 511 | 6.5 | 21.9 |
| 20 | LD5-1 | Lake Pt E, 1m | 511 | 6.4 | 20.2 |
| 21 | LD5-2 | Lake Pt E, 2m | 512 | 6.4 | 24.2 |
| 22 | LD5-3 | Lake Pt E, 3m | 519 | 6.0 | 22.4 |
| 23 | LD5-4 | Lake Pt E, 4m | 557 | 6.2 | 21.7 |
| **Groundwater sources** | | | | | |
| 24 | PPBH | Patandi Borehole(↑) | 468 | -2.4 | -13.3 |
| 25 | MHBH | Meru H. Borehole(↑) | 376 | -1.9 | -14.3 |
| 26 | MASP | Makisoro Spring(↑) | 364 | -2.1 | -12.9 |
| 27 | CASP | Carmatec Spring(↑) | 196 | -2.2 | -13.9 |
| 28 | SASP | Saibala Spring(↑) | 277 | -2.2 | -15.0 |
| 29 | TBB1 | Tengeru Borehole 1(↑) | 462 | -1.0 | -8.3 |
| 30 | TBB2 | Tengeru Borehole 2(↑) | 466 | -1.3 | -8.4 |
| 31 | RCBH | Roman Borehole(↑) | 468 | -2.1 | -10.4 |
| 32 | DSBH | Duluti S. Borehole(↑) | 435 | -2.2 | -11.6 |
| 33 | LTSP | Lita Spring(↓) | 402 | -1.6 | -13.5 |
| 34 | NMB1 | NM Borehole 1(↓) | 830 | -1.1 | -4.8 |
| 35 | NMB2 | NM Borehole 2(↓) | 888 | -1.7 | -3.9 |
| 36 | CDBH | CDTI Borehole(↓) | 682 | -2.0 | -10.9 |
| 37 | CDSP | CDTI spring(↓) | 565 | -2.5 | -11.7 |





| 38 | Rain | Close to the lake | -1.2 | -1.6 |


**Figure 1: Study area showing the geology of the area and sampling site in Lake Duluti and surrounding water sources.**








**Figure 2: Seasonal variations (dry and wet seasons) in pH (a, b) and Temperature (c, d) with depth for Lake Duluti.**



**Figure 3: Variation of Sodium with depth for Lake Duluti during (a) dry and (b) wet season.**

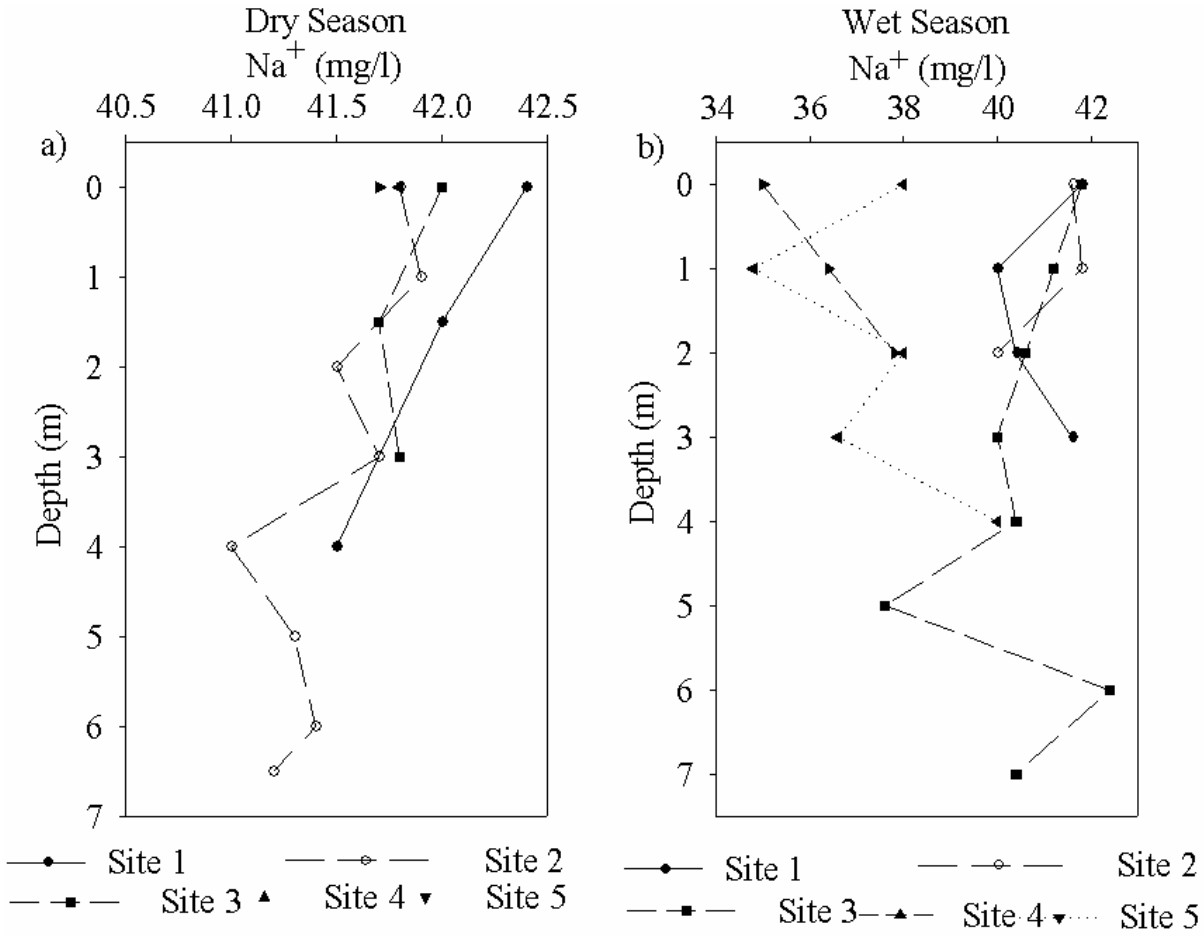






**Figure 4: Piper trilinear diagram for Facies Classification in dry season.**

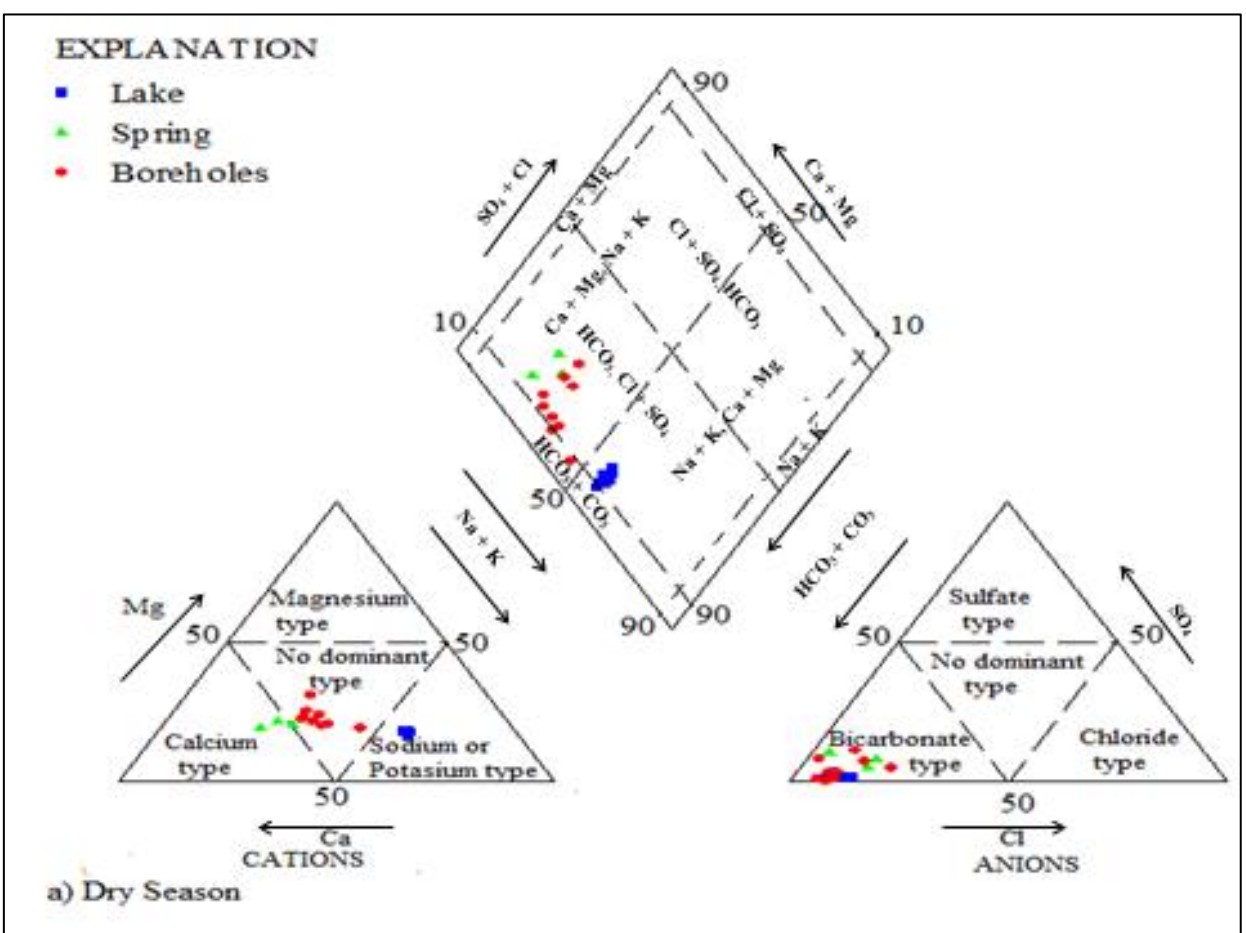


















**Figure 5: Piper trilinear diagram for Facies Classification in wet season**

















**Figure 6: Plot of $\delta^{18}O$ Vs $\delta^{2}H$ of water samples with respect to the Local Meteoric Water Line (LMWL).**

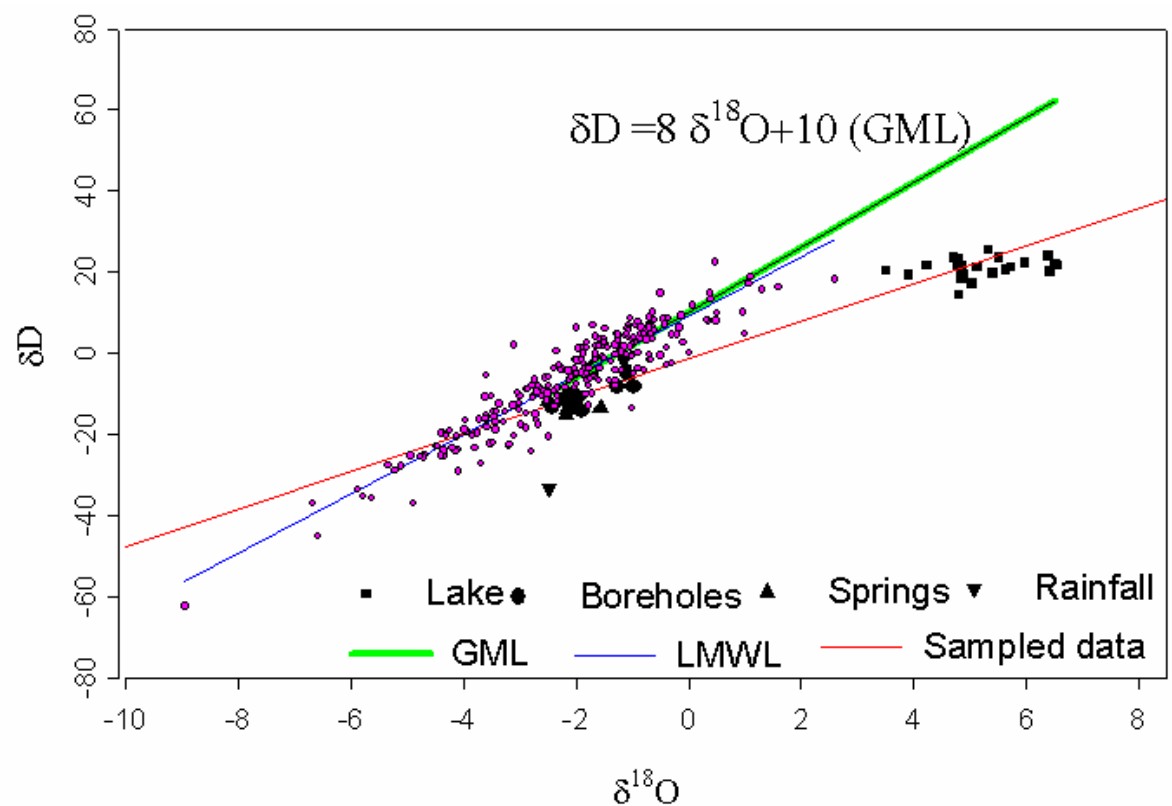


















**Figure 7: Trend of oxygen isotopic composition with depth in Lake Duluti**