# Peer review of "APPLICATION OF ISOTOPES AND WATER BALANCE ON"

_Hydrology and Earth System Sciences, 2016_

## Referee Comment (RC1) · Anonymous Referee #1 · 5 Jul 2016

Referee report on HESS 2016-176: Application of Isotopes and Water Balance on Lake Duluti-Groundwater Interaction, Arusha, Tanzania By Mduma et al.

The manuscript uses tracer data (major chemistry and stable isotopes) to characterize the surface-groundwater exchange and the water balance of a specific Lake in Tanzania. Unfortunately the manuscript is characterized by several major shortcomings that do not warrant publication in a scientific journal. The reasons are detailed down below.

1. Lacking scientific background

The scientific background disregards existing work in the field, is much too general and not to the point of the actual study. It only contains local investigations (many of

them also in grey literature) and most of them in groundwater. There is virtually no study on a lake water balance or on lake-groundwater exchange included that would proves that the authors are aware of the actual methodology to be used. Starting with the pioneering works of Joel Gat there are many examples in the literature that used environmental isotopes to characterize lake water balances and also the exchange with surrounding groundwater.

2.Wrong assumptions in the isotopic mass balance

As a consequence of a missing scientific background there are major shortcomings in the applied methods that produce totally wrong results. The most alarming mistake is included in equation 7: It is simply wrong that the isotopic composition of lake water can be derived by simply adding isotopic concentrations of inflow, precipitation and evaporation. It is known from literature that the isotopic enrichment of an evaporating water body is a complex process and depends on various factors. The fundamental equations are given in various scientific papers, with Gat & Browser (1991) being only one example. And even if a traditional mixing approach was applicable (this could eventually be for other tracers, e.g. major ions), tracer concentrations would have to be weighted by the amounts of the different components.

3. Uncertain laboratory procedures for major ions

It is acknowledged that the authors are from a developing country with limited laboratory resources. However, the applied methods are only briefly described and no estimation on the error of the procedures is possible. Which "multi-parameter meter" was used for the onsite field measurements? Why were some major ions measured by a multi-parameter spectrometer (K, No3, So4) and others by tritration (Ca, Mg, Co3, Cl) and Na by a flame photometer?

4. Non-necessary freezing of isotope samples

Sampels for stable water isotopes are stable for many years if they are filled without

headspace and kept in tight bottles. Why were the samples for isotopes frozen in glass bottles? How was breakage prevented during freezing? How was tightness guaranteed during freezing and volume expansion? In the freezer a leakage will cause sublimation and additional fractionation. If studies like this are published, other researchers will perhaps freeze their samples too which causes non-necessary problems for them.

5. Violated assumption of complete mixing of the lake water body

The authors admit themselves that mixing inside the lake was poor, because concentrations varied between different locations and different depths. Nevertheless they assume complete mixing and used single mean values for the lake water balance. for this they form averages of various samples collected in the dry and wet season. But fractionation by evaporation is higher at the lake surface and groundwater inflow occurs at certain depths only. In addition, no exact sampling dates are given, it seems that samples were averaged arbitrarily, when is a wet season sample a wet season sample?

6. Poor interpretation of tracer data

The interpretation of the measured major ion chemistry is not convincing. Only for some ions it is argued that concentration in the lake is higher than in groundwaters due to evaporation, but this is principally true for all ions. Also anthropogenic inputs are only related to high $SO_4$ not to other ions. A positive correlation of $SO_4$ with $NO_3$ is no indication for oxidation of organic matter, there may be many other factors playing a role here, primary production inside the lake is only one. But also the isotopic data interpretation of figure 6 is limited: First of all a straight line through all sampled data does not make sense, because different types are mixed. Second, the "local meteoric water line" stems from samples virtually sampled at many different locations across Tanzania. They produce a very large scatter and cannot be related to the samples of the present study.

---

## Referee Comment (RC2) · Anonymous Referee #2 · 7 Jul 2016

General comments

The manuscript "Application of isotopes and water balance on Lake Duluti–groundwater interaction, Arusha, Tanzania" by Mduma et al. uses major ion and stable isotope data to assess lake water – groundwater interactions in a volcanic area of northern Tanzania. This study presents an important dataset from a data scarce region, and as such it would be of value for the local stakeholders. However, in its current form the manuscript is a mere case study with little global significance, and more importantly it suffers from several major flaws that, in my opinion, prevent it from being published in HESS. In particular, the introduction fails in properly addressing the research background; the water balance calculations are based on incorrect assumptions; the results are poorly presented, confusing and they lack a link between the sections on major ion chemistry and the sections on water balance calculations. There are also many grammatical and typesetting errors all throughout the manuscript, and figures are of poor quality. I would encourage the authors to thoroughly rework each section of their manuscript and to be much more meticulous in their assumptions, presentation of results, analyses and interpretations. Specific details are given below.

Specific comments

Introduction

* The Introduction does not establish the context of the investigation properly. The research objectives need to be clarified, and the authors should strive to go beyond the simple case study by addressing broader questions that will be relevant to the whole hydrology research community. Despite being a well-covered topic, the authors do not provide a single reference on the interactions between lake water and groundwater. Just a few suggestions: Krabbenhoft et al. (1990), Gibson and Edwards (1996), Kebede et al. (2006) (and several other papers in Ethiopia), Bouchez et al. (2016). A more stringent literature review is needed here, and only from the knowledge gaps thus identified should the authors derive their research questions.

* The authors state that their objective is to "quantify the groundwater exchange with Lake Duluti" (L.58-59), but this objective does not seem to cover all the Results section. The water types and hydrochemical processes occurring in each hydrological component are also described, which needs to be reflected in the research questions presented in the Introduction.

Methods

* The structure of the Methods section could do with some adjustments. I would recommend adding two subsections '2.1 Sampling and laboratory analyses' (L.81) and '2.2 Water balance approach' (L.111). As it is currently written, subsection 2.1 goes

back and forth between ionic and isotopic analyses, which tends to confuse the reader. This needs to be improved.

* The reported analytical errors for deuterium and oxygen-18 are extremely high compared to most studies. Any comment on why this may be? Error bars should be added to the graphs in Figure 7 (the observed spatial variations may not be significant).

* I see several major flaws to the isotopic mass balance and mixing methods developed by the authors. First, equation (7) does not introduce weighting factors for each component, so the equation is incorrect. Second, the mass balance presented in equations (9) and (10) has two unknown terms, i.e. groundwater inflow Gi and groundwater outflow Go (or it is unclear to me whether one of the two terms is known, and how it is estimated); therefore the mass balance cannot be resolved using only one equation. Third, the mixing approach assumes that the two end-members are (1) lake water and (2) groundwater. Surprisingly, the mixed component to be determined also seems to be lake water. How can a single term be both the mixed component and an end-member in the calculation?

Results and Discussion

* I would suggest revising the structure of the Results and Discussion section. In particular, it would be more convenient for the reader to first be presented with the hydrochemical facies of each component. Please consider moving subsection 3.1.2 at the beginning of section 3 (also note that for some reason subsection 3.1.1 is missing in the current manuscript).

* The Piper diagrams shown in Figures 4 and 5 may be dispensable, unless they are used to convey further information such as the delineation of hydrochemical processes, mixing between different components, etc. The observed changes from dry to wet are not discussed in the text. All other figures are not sufficiently described, some being not even referred to in the text (e.g. Figure 2).

* Generally, it is not clear how the seasonality factor is addressed. Can the authors elaborate on the effect of seasonality on major ion chemistry? And on the water balance? Also, the dry sampling round was carried out in January–February (L.83), however the authors mentioned that rainfall in the region occurs from October–November to May. Please elaborate on the weather conditions for that specific year.

* The way the end-member mixing analysis has been undertaken and the way the results are reported are both extremely confusing to me. It is unclear how the end-members were defined, and a proper characterisation of each end-member in terms of water types and isotopic ratios is clearly missing. This should be a prerequisite to carrying out any mixing analysis. Importantly, how do the seasonal changes affect your mixing analysis? How was this handled? Furthermore, the indication that "the fraction of groundwater is 0.73" (L.293) comes out of the blue and comes without any form of justification. How was this calculated?

* As per a previous comment, it is unclear to me what the unknowns in the mass balance equation are. I suspect that the authors had to constrain either $G_i$ or $G_o$? If so, how was this done? How did you obtain the values 2,430,960 m3/y (L.279) and 2,902,620 m3/y (L.280)? Where does the "80%" value (L.295) come from? Also, on which basis can you state that 73% and 80% are significantly different?

* The authors could have used the isotopically-enriched, highly evaporated lake water signature as a tracer of lake water inflow into the underlying aquifer. Do the major ion and isotopic values match your hypothesis of water loss from the lake to the aquifer? Are there different signatures upgradient vs downgradient the aquifer? Once again too many pieces of information are missing.

Technical corrections

I will not go through technical corrections because they are too many at this stage, and the paper first requires proofreading by the authors themselves. However, I would agree to assess a revised version of the manuscript if necessary.
Cited references

Bouchez, C., Goncalves, J., Deschamps, P., Vallet-Coulomb, C., Hamelin, B., Doum-nang, J.-C., Sylvestre, F., 2016. Hydrological, chemical, and isotopic budgets of Lake Chad: a quantitative assessment of evaporation, transpiration and infiltration fluxes. Hydrol. Earth Syst. Sci. 20, 1599–1619.

Gibson, J.J., Edwards, T.W.D., 1996. Development and validation of an isotopic method for estimating lake evaporation. Hydrol. Process. 10, 1369–1382.

Kebede, S., Travi, Y., Alemayehu, T., Marc, V., 2006. Water balance of Lake Tana and its sensitivity to fluctuations in rainfall, Blue Nile basin, Ethiopia. J. Hydrol. 316, 233–247.

Krabbenhoft, D.P., Bowser, C.J., Anderson, M.P., Valley, J.W., 1990. Estimating ground-water exchange with lakes: 1. The stable isotope mass balance method. Water Resour. Res. 26, 2445–2453.

---

## Referee Comment (RC3) · Anonymous Referee #3 · 1 Aug 2016

This manuscript presents a hydrochemical data set (stable water isotopes and major ions) to evaluate interactions between in- and out-flows of Lake Duluti (Tanzania). The authors present an extensive dataset from a relatively data sparse region. Hence, once collected and analysed systematically and comprehensively, this work could make an important contribution. Unfortunately, the work is not presented well and I would not recommend the manuscript for publication in its current form.

Firstly, the work is currently presented as a case study which might be relevant for the local area only. The authors do not recognise the vast amount of scientific literature on lake surface water groundwater interactions (including the use of tracers) and do not clearly indicate what the relevance and wider implications are of this particular study

for the wider scientific community.

Secondly, the manuscript is quite difficult to follow owing to poor English, unclear structure and poor quality of the tables and figures (and inappropriate references in the text, e.g. where is Figure 0 or Table 6?). I would recommend to split the results and discussion section so that all results are presented clearly and systematically first. The discussion should then focus on the interpretation of the dataset as a whole. It would also be helpful if there were clear objectives or specific research questions that the discussion could be framed around.

There are also several methodological shortcomings as already pointed out by the other reviewers including some of the data collection procedures and the assumptions in the mass balance calculations. It was also not clear if precipitation was sampled locally during this study. The hydro-geochemical facies section presents the data separately for two seasons, but a clear interpretation that describes the processes underlying the different patterns is missing.

A detailed list of specific comments is not provided as the authors need to address the major issues as addressed above first. It might also be helpful if the manuscript could be proofread before re-submission.

Please also consider revising the title as it does not flow well and does not represent the full contents of the paper well.

---

## Referee Comment (RC4) · Anonymous Referee #4 · 3 Aug 2016

Dear Editor, dear authors,

there is not much to add to the reviews given before: The paper presents an interesting data set for a data scarce region, but the way, how this data set is presented is problematic (methodological errors, missing literature background). Thus, I follow the other reviewers and do not recommend the publication of the paper in it's present form.

What strikes me most about this paper, is on the one hand the presentation of a huge set of equations, which are either unnecessary (e.g. equation 3 and 4, 5 and 6, 9 and 10_1 are exactly the same equations) or wrong (e. g. equation 7, as mentioned by the other reviewers) or contradicting (eq. 12 uses the fraction of lake water (fL) to define $\delta$mix, while eq. 13 uses $\delta$mix for the calculation of fL) and on the other hand the results

of these calculations do not appear in the paper in a quantitative/ comparative way at all.

I think, this study could be an interesting paper, if the previous recommendations by the other reviewers would be considered (amongst others a detailed description of lake water colomn mixing processes should be provided) and if a) all boundary conditions of the local setting of the lake would be strictly analyzed and defined (and presented in the paper), b) the set of equations would be set properly (by considering the literature recommended by the reviewers 1 and 2) and c) if there would be a clear presentation of the quantitative results of the water balance calculations together with an interpretation/discussion what these results imply for the local aquifers and their management in a much more specific/quantitative way (e.g. what is the main gw flow direction in the study area and how/where does the lake system changes water availablity?).

In summary, I recommend rejection, with the possibility to resubmit after implementing all required changes.

---

## Author Comment (AC1) · 1 Sep 2016

Response to Anonymous Reviewers

Please refer to the interactive comments of anonymous reviewers for a manuscript entitled "Application of isotopes and water balance on Lake Duluti–groundwater interaction, Arusha, Tanzania" by N. P. Mduma et al. We thank the reviews of valuable comments, and suggested areas for improvement. We will use the critical comments to improve and strengthen the analysis of our manuscript. We agree with the reviewers that the paper needs to be strengthened. Given another opportunity to improve the manuscript, the introduction section will be re-written to include current scientific understanding in the field (rigorous literature review on lake water balance, lake-groundwater interaction), define clearly the problem (knowledge gap), and objectives. We will revisit all the assumptions made in formulating all equations and all errors made and unnecessary equations such as equation 7 will be rectified using well-developed approaches. This applies to the first two comments of the first reviewer shown below. Also because comments from the four reviewers are similar response to the comments from the first reviewer general covers for the others. We will do our level best to ensure that there are no grammatical and typographic errors.

1. Lacking scientific background The scientific background disregards existing work in the field, is much too general and not to the point of the actual study. It only contains local investigations (many of them also in grey literature) and most of them in groundwater. There is virtually no study on a lake water balance or on lake-groundwater exchange included that would proves that the authors are aware of the actual methodology to be used. Starting with the pioneering works of Joel Gat there are many examples in the literature that used environmental isotopes to characterize lake water balances and also the exchange with surrounding groundwater.

Comments 2.Wrong assumptions in the isotopic mass balance As a consequence of a missing scientific background there are major shortcomings in the applied methods that produce totally wrong results. The most alarming mistake is included in equation 7: It is simply wrong that the isotopic composition of lake water can be derived by simply adding isotopic concentrations of inflow, precipitation and evaporation. It is known from literature that the isotopic enrichment of an evaporating water body is a complex process and depends on various factors. The fundamental equations are given in various scientific papers, with Gat & Browser (1991) being only one example. And even if a traditional mixing approach was applicable (this could eventually be for other tracers, e.g. major ions), tracer concentrations would have to be weighted by the amounts of the different components.

Comments 3. Uncertain laboratory procedures for major ions It is acknowledged that the authors are from a developing country with limited laboratory resources. However,

the applied methods are only briefly described and no estimation on the error of the procedures is possible. Which "multi-parameter meter" was used for the onsite field measurements? Why were some major ions measured by a multi-parameter spectrometer (K, No3, So4) and others by tritration (Ca, Mg, CO3, Cl) and Na by a flame photometer? Authors' response It is indeed true that laboratory resources are often limited in the developing south. We have a used a range of methods to overcome some of these challenges. For in-situ measurement we used HANNA Multi-Parameter instrument. We will improve on the description of the methods applied. Methodology Comment from Reviewer 1 4. Non-necessary freezing of isotope samples Samples for stable water isotopes are stable for many years if they are filled without headspace and kept in tight bottles. Why were the samples for isotopes frozen in glass bottles? How was breakage prevented during freezing? How was tightness guaranteed during freezing and volume expansion? In the freezer a leakage will cause sublimation and additional fractionation. If studies like this are published, other researchers will perhaps freeze their samples too which causes non-necessary problems for them.

Response: Samples were not frozen but rather were stored in a refrigerator. This was a typing error as the samples were stored in a refrigerator and not freezer. Thanks you for pointing us to this statement, we will modify accordingly. 5. Violated assumption of complete mixing of the lake water body The authors admit themselves that mixing inside the lake was poor, because concentrations varied between different locations and different depths. Nevertheless they assume complete mixing and used single mean values for the lake water balance. For this they form averages of various samples collected in the dry and wet season. But fractionation by evaporation is higher at the lake surface and groundwater inflow occurs at certain depths only. In addition, no exact sampling dates are given, it seems that samples were averaged arbitrarily, when is a wet season sample a wet season sample?

Northern Tanzania is characterized by two main rain seasons (bimodal rainfall) namely the long rains and the short rains, which are associated with the northward and southward migration of the Inter-Tropical Convergence Zone (ITCZ), respectively (Kabanda and Jury, 1999; Zorita and Tilya, 2002; Kijazi and Reason 2009). The long rains begin in the mid of March and end at the end of May, while the short rains begin in the middle of October and continues to early December. Fieldwork for the dry season was conducted between January and February 2015, while that of wet season was conducted between the months of March and April, 2015. Dates of sampling will be provided in revised version of the manuscript. Assumptions made about mixing in the lake will be reviewed as we did contradict ourselves.

6. Poor interpretation of tracer data The interpretation of the measured major ion chemistry is not convincing. Only for some ions it is argued that concentration in the lake is higher than in groundwaters due to evaporation, but this is principally true for all ions. Also anthropogenic inputs are only related to high SO4 not to other ions. A positive correlation of So4 with NO3 is no indication for oxidation of organic matter, there may be many other factors playing a role here, primary production inside the lake is only one. But also the isotopic data interpretation of figure 6 is limited: First of all a straight line through all sampled data does not make sense, because different types are mixed. Second, the "local meteoric water line" stems from samples virtually sampled at many different locations across Tanzania. They produce a very large scatter and cannot be related to the samples of the present study.

We highly acknowledge for pointing out weaknesses in interpretation of our data. We will re-analyze our data and provide a comprehensive data interpretation. Availability of data to reconstruct a local meteoric water line for Arusha region only is not available and our idea was to construct a local meteoric water line for Tanzania by utilizing isotope data available at the GNIP website, actual measurements of precipitation for samples collected during this study and other reported data elsewhere, and then compare it with the results of the present study.

References Kabanda, T.A. and. Jury, M.R., 1999. Inter-annual variability of short rains over northern Tanzania. Climate Research 13, 231–241. Kijazi, A.L. and Reason,

C.J.C., 2009. Analysis of the 2006 floods over northern Tanzania. International Journal of Climatology 29, 955–970. Zorita, E., and Tilya, F.F., 2002. Rainfall variability in Northern Tanzania in the March–May season (long rains) and its links to March–May season (long rains) and its links to large-scale climate forcing large-scale climate forcing. Climate Research 20, 31–40.
* * *
Interactive
comment